# Characterization of intracellular calcium mobilization induced by remimazolam, a newly approved intravenous anesthetic

Tomoaki Urabe[1], Hirotsugu Miyoshi[1]*, Soshi Narasaki[1], Yuhki Yanase[2], Kazue Uchida[3], Soma Noguchi[4], Michihiro Hide[3¤], Yasuo M. Tsutsumi[1], Norio Sakai[4]

**1** Department of Anesthesiology and Critical Care, Graduate School of Biomedical & Health Sciences, Hiroshima University, Hiroshima, Japan, **2** Department of Pharmacotherapy, Graduate School of Biomedical and Health Sciences, Hiroshima University, Hiroshima, Japan, **3** Department of Dermatology, Graduate School of Biomedical and Health Sciences, Hiroshima University, Hiroshima, Japan, **4** Department of Molecular and Pharmacological Neuroscience, Graduate School of Biomedical & Health Sciences, Hiroshima University, Hiroshima, Japan

¤ Current address: Department of Dermatology, Hiroshima City Hiroshima Citizens Hospital, Hiroshima, Japan

* h-miyoshi@hiroshima-u.ac.jp

**Data Availability Statement:** All relevant data are within the manuscript and its Supporting information files.

## Abstract

Many anesthetics, including Propofol, have been reported to induce elevation of intracellular calcium, and we were interested to investigate the possible contribution of calcium elevation to the mechanism of the newly approved remimazolam actions. Remimazolam is an intravenous anesthetic first approved in Japan in July 2020, and is thought to exert its anesthetic actions *via* γ-aminobutyric acid A (GABA$_A$) receptors; however, the precise mechanisms of how remimazolam elevates intracellular calcium levels remains unclear. We examined the remimazolam-induced elevation of intracellular calcium using SHSY-5Y neuroblastoma cells, COS-7 cells, HEK293 cells, HeLa cells, and human umbilical vein endothelial cells (HUVECs) loaded with fluorescent dyes for live imaging. We confirmed that high concentrations of remimazolam (greater than 300 μM) elevated intracellular calcium in a dose-dependent manner in these cells tested. This phenomenon was not influenced by elimination of extracellular calcium. The calcium elevation was abolished when intracellular or intraendoplasmic reticulum (ER) calcium was depleted by BAPTA-AM or thapsigargin, respectively, suggesting that calcium was mobilized from the ER. Inhibitors of G-protein coupled receptors (GPCRs)-mediated signals, including U-73122, a phospholipase C (PLC) inhibitor and xestospongin C, an inositol 1,4,5-triphosphate receptors (IP$_3$R) antagonist, significantly suppressed remimazolam-induced calcium elevation, whereas dantrolene, a ryanodine receptor antagonist, did not influence remimazolam-induced calcium elevation. Meanwhile, live imaging of ER during remimazolam stimulation using ER-tracker showed no morphological changes. These results suggest that high doses of remimazolam increased intracellular calcium concentration in a dose-dependent manner in each cell tested, which was predicted to be caused by calcium mobilization from the ER. In addition, our studies using various inhibitors revealed that this calcium elevation

**Funding:** This study was supported by a Grant-in-Aid for Scientific Research from the Ministry of Education, Sports, and Culture (21K16560, 19H03409, 21K08948, 19K09353), and by grants from the Takeda Science Foundation, the Uehara Memorial Foundation, and the Japanese Smoking Research Association. The funders had no role in study design, data collection and analysis, decision to publish, or preparation of the manuscript.

**Competing interests:** The authors have declared that no competing interests exist.

might be mediated by the GPCRs-IP$_3$ pathway. However, further studies are required to identify which type of GPCRs is involved.

## 1. Introduction

Remimazolam is a medication for the induction and maintenance of general anesthesia like propofol. Remimazolam is from the same pharmacological class as the existing midazolam (category- benzodiazepines) and was approved for medical use in Japan in July 2020. Remimazolam is approved for procedural sedation in US, Europe and China and for general anesthesia in South Korea, and it is likely to be used more widely in the future. It has the same advantages as midazolam, such as less circulatory inhibitory effect, less injection site reaction at the time of administration, and is antagonized by flumazenil. Remimazolam acts on GABA$_A$ receptors, but the mechanisms underlying its side effects are unclear [1, 2]. In general, anesthetic-induced elevation of intracellular calcium has been implicated in the development of a variety of side effects [3–5]. We have shown that propofol acts directly on intracellular organelles, such as the endoplasmic reticulum (ER) to elevate intracellular calcium [6]. This phenomenon may be involved in the development of excessive hypotension and propofol-induced vascular pain. Propofol-induced calcium elevation may induce vasodilation by activating the intracellular signaling pathway and promoting the phosphorylation of NO synthase, resulting in the synthesis of NO. This may contribute to hypotension and vascular pain [6]. In our recent study, we also found that high concentrations of remimazolam increase the intracellular calcium [7]. The aim of this study was to elucidate the mechanism underlying remimazolam-induced intracellular calcium elevation.

## 2. Materials and methods

### 2.1. Materials

Remimazolam was kindly provided from Mundipharma Japan and PAION Deutschland GmbH. Caffeine, dantrolene, and thapsigargin were obtained from FUJIFILM Wako Pure Chemical Industries, Ltd. (Osaka, Japan). Acetylcholine was purchased from Sigma-Aldrich (St. Louis, Massachusetts). U-73122 and xestospongin C were purchased from Cayman Chemical Company (Ann Arbor, Michigan). ER-Tracker Red and Fluo-4 were purchased from Molecular Probes (Eugene, Oregon). Glass-bottom culture dishes were purchased from Mat-Tek Corporation (Ashland, Oregon). All other chemicals used were of analytical grade.

### 2.2. Cell culture

SHSY-5Y cells and HEK293 cells were purchased from Riken Cell Bank (Tsukuba, Japan). Human umbilical vein endothelial cells (HUVECs) were obtained from American Type Culture Collection (ATCC, Manassas, Virginia). SHSY-5Y cells were maintained in Dulbecco's modified Eagle's medium/Ham's F-12 medium (FUJIFILM Wako, Osaka, Japan). HEK293 cells were cultured in Dulbecco's modified Eagle's medium (FUJIFILM Wako, Osaka, Japan). The medium for SHSY-5Y and HEK293 cells were supplemented with 10% fetal bovine serum, penicillin (100 units/mL), and streptomycin (100 μg/mL). HUVECs were cultured as previously described [8]. Cell cultures were maintained in a humidified atmosphere containing 5% CO$_2$ at 37 ˚C in the dark. Cells were seeded in glass-bottomed culture dishes 1–2 days before imaging.

### 2.3. Loading of Fluo-4 and ER-Tracker Red

The culture medium of the cells was replaced with normal HEPES buffer composed of 165 mM NaCl, 5 mM KCl, 1 mM $MgCl_2$, 1 mM $CaCl_2$, 5 mM HEPES, and 10 mM glucose, pH 7.4. Then, the cells were incubated with calcium indicator Fluo-4 (125 μg/mL) and ER-Tracker Red (1 μM) under light shielding at 37 ˚C for 15–20 min prior to microscopic observations. In the experiments using HUVECs and COS-7 cells, Pluronic F-127 was also used to facilitate the loading of Fluo-4 and ER-Tracker Red.

### 2.4. Drug treatment

Remimazolam besylate was diluted in dimethyl sulfoxide (DMSO) to prepare a 100 mM stock solution. The same DMSO concentration was used as the negative control. Remimazolam or DMSO was diluted to the desired concentrations with normal HEPES buffer and was applied to the cells at the appropriate final concentrations. Sonication was performed immediately before the addition of remimazolam. Treatment concentrations and durations are described in each figure and caption. U-73122, xestospongin C, and dantrolene were applied 15 minutes before the treatment with remimazolam.

### 2.5. Observation of intracellular calcium elevation and morphological changes in intracellular organelles

Fluorescence images were taken with a BZ-9000 fluorescent microscope (KEYENCE, Osaka, Japan). We used GFP-B (KEYENCE, excitation wavelength 470/40 nm, emission wavelength 535/50 nm) as the excitation and emission filters and S Fluor X40/0.93 (NIKON) as the objective lens. To visualize remimazolam-triggered calcium elevation and morphological changes in the ER, time-lapse imaging at 15-second intervals was performed after treatment of the cells with these drugs.

### 2.6. Semiquantitative evaluation of remimazolam-induced intracellular calcium elevation

The time course of typical remimazolam-induced intracellular calcium elevation is shown in Fig 1, which was evaluated according to a previously described method [6].

In Fig 1, fluorescence intensity in the circled area was used as background. The red and blue lines indicate the average fluorescence changes in cells in the total area and background, respectively. The differences between both lines before each drug application (b) and at the peak of calcium elevation (a) were measured. The a to b ratio (a/b) was considered as an index of remimazolam-induced calcium elevation.

### 2.7. Statistical analysis

Prism 4 software (GraphPad Software, San Diego, CA) was used to conduct statistical analyses. Statistical significance was determined by one-way ANOVA, followed by Dunnett's post-test or Mann Whitney test. All data represent the mean ± SEM (standard error of the mean). Differences were considered significant when the p value was less than 0.05 ($p < 0.05$).

## 3. Results

### 3.1. Characterization of remimazolam-induced calcium elevation

We observed a dose-dependent remimazolam-induced elevation of intracellular calcium at concentrations between 0 and 500 μM in SHSY-5Y cells and HUVECs (Fig 2A and 2B).

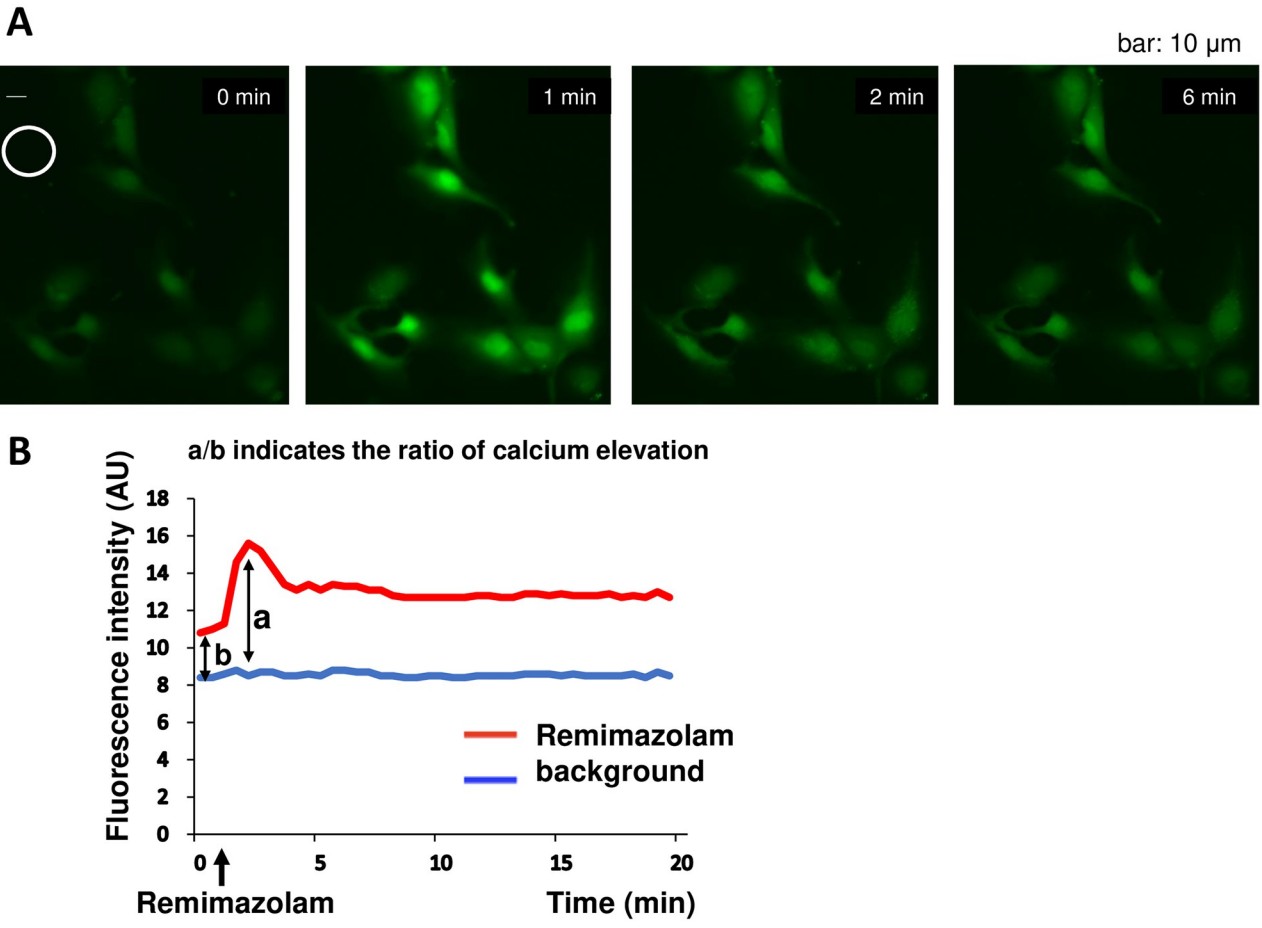

**Fig 1. Remimazolam-induced intracellular calcium elevation.** (A) An example of a typical remimazolam-induced intracellular calcium elevation time series in HUVECs. Approximately one min after the administration of 300 μM remimazolam, an increase in the fluorescence intensity of Fluo-4 was observed, indicating that intracellular calcium was increased. Fluorescence intensity in the circled area was used as background. (B) The method of semi-quantifying the rate of calcium elevation. The red line indicates the average fluorescence change over the entire observed region. The blue line indicates the fluorescence change in the background region without cells. The differences between both lines before the remimazolam application (b) and at the peak of calcium elevation (a) were measured. The ratio a to b (a/b) was considered as an index of remimazolam-induced calcium elevation.

Remimazolam at a concentration greater than or equal to 300 μM significantly induced the elevation of intracellular calcium in a dose-dependent manner in both cell lines. In addition, remimazolam-induced intracellular calcium elevation was observed in HEK293 cells (S1 Fig).

## 3.2. Remimazolam-induced calcium elevation in the absence of extracellular calcium

To elucidate whether the elevation of intracellular calcium was mediated by calcium influx from the extracellular buffer, we eliminated calcium from the extracellular buffer and observed a remimazolam-induced elevation of intracellular calcium at concentrations between 0 and 500 μM in SHSY-5Y cells and HUVECs. Remimazolam at a concentration greater than or equal to 300 μM significantly induced a dose-dependent increase in intracellular calcium in SHSY-5Y cells (Fig 3A). Remimazolam at 500 μM also significantly induced a dose-dependent increase in intracellular calcium levels in HUVECs (Fig 3B).

## Calcium-containing external solution

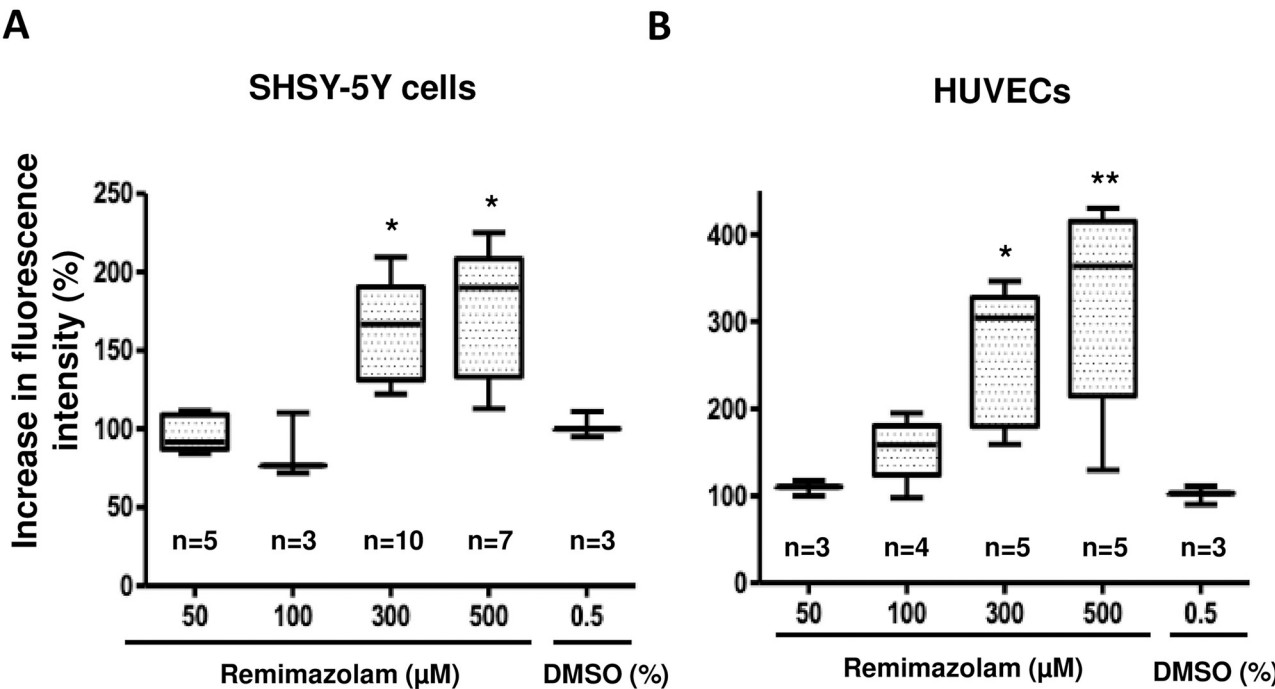

**Fig 2. Remimazolam-induced intracellular calcium elevation in SHSY-5Y cells and HUVECs.** (A) Remimazolam at a concentration greater than or equal to 300 μM significantly induced the elevation of intracellular calcium in a dose-dependent manner in SHSY-5Y cells (n = 3–7, * $p < 0.05$, compared to control, one-way ANOVA followed by Dunnett's post-test). The horizontal line in each box indicates the median, the box shows the interquartile range (IQR), and the whiskers represent 1.5 × IQR. (B) Remimazolam at a concentration greater than or equal to 300 μM significantly induced the elevation of intracellular calcium in a dose-dependent manner in HUVECs (n = 3–5, * $p < 0.05$, ** $p < 0.01$, compared to control, one-way ANOVA followed by Dunnett's post-test). The horizontal line in each box indicates the median, the box shows the interquartile range (IQR), and the whiskers represent 1.5 × IQR.

These results are very similar to those seen in Fig 2A and 2B, suggesting that the remimazolam-induced elevation of intracellular calcium levels may not be influenced by the elimination of extracellular calcium.

### 3.3. Remimazolam-induced calcium elevation in the absence of intracellular calcium

To elucidate the influence of intracellular calcium levels on remimazolam-induced calcium elevation, we treated the cells with BAPTA-AM (20 μM), a calcium chelator useful for manipulating intracellular free calcium levels. BAPTA-AM significantly reduced and almost abolished the remimazolam (300 μM)-induced calcium elevation in both SHSY-5Y cells and HUVECs (Fig 4A and 4B).

Similar effects of BAPTA-AM were observed even with 500 μM remimazolam (Fig 4A and 4B).

### 3.4. Effects of thapsigargin on remimazolam-induced calcium elevation

To elucidate whether remimazolam-induced calcium elevation was mobilized from the ER, we investigated the effect of thapsigargin (TG), a $Ca^{2+}$-ATPase inhibitor that eliminates calcium from the ER. As shown in Fig 5, after treatment with 5 μM TG (①), $[Ca^{2+}]_i$ was elevated.

## Calcium-free external solution

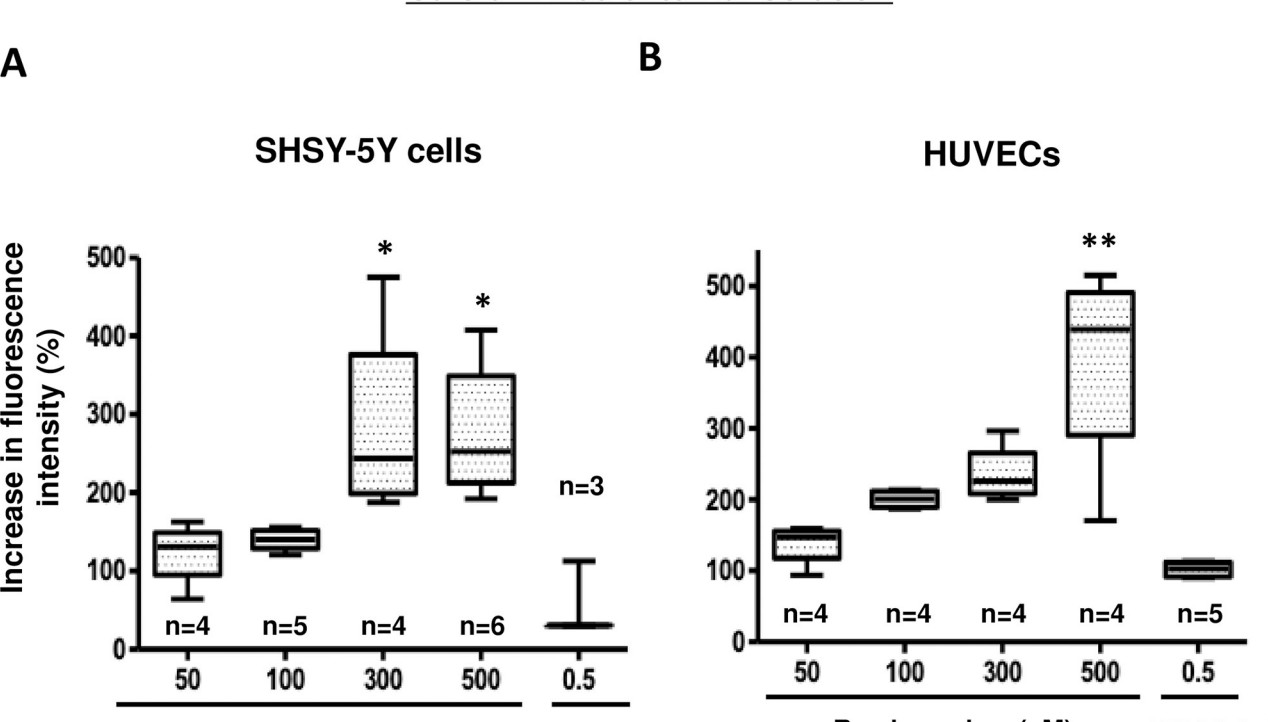

**Fig 3. Characterization of remimazolam-induced intracellular calcium elevation in the absence of extracellular calcium.** After eliminating calcium from the extracellular buffer, remimazolam-induced elevation of intracellular calcium was observed at concentrations between 0 and 500 μM in SHSY-5Y cells and HUVECs. (A) Remimazolam (300 μM)-induced elevation of intracellular calcium was not influenced by the elimination of extracellular calcium in SHSY-5Y cells (n = 3–7, * $p < 0.05$, compared to control, one-way ANOVA followed by Dunnett's post-test). The same concentration of DMSO corresponding to 500 μM of remimazolam was administered as a control. The horizontal line in each box indicates the median, the box shows the interquartile range (IQR), and the whiskers represent $1.5 \times$ IQR. (B) Remimazolam (300 μM)-induced elevation of intracellular calcium was not influenced by the elimination of extracellular calcium in HUVECs (n = 3–7, * $p < 0.05$, compared to control, one-way ANOVA followed by Dunnett's post-test). The same concentration of DMSO corresponding to 500 μM of remimazolam was administered as a control. The horizontal line in each box indicates the median, the box shows the interquartile range (IQR), and the whiskers represent $1.5 \times$ IQR.

Remimazolam was added at a concentration of 300 μM 30 min after TG administration, when the $[Ca^{2+}]_i$ had almost returned to the basal level (②). Remimazolam did not induce any calcium elevation, suggesting that remimazolam mobilized calcium from the ER.

### 3.5. Observation of remimazolam-induced morphological changes in the ER by ER-Tracker Red

We found that propofol alters the morphology of the ER, likely by fragmentation or aggregation, and causes calcium leakage into the cells. Therefore, we investigated the effects of remimazolam on ER morphology. ER-Tracker Red dye, is a cell-permeant marker capable of staining the ER in living cells with high selectivity. Cells were stained with this dye before remimazolam treatment, and ER status was observed during remimazolam stimulation. In this study, we did not see any morphological changes in the ER of both SHSY-5Y cells and HUVECs after the administration of remimazolam at any concentration (S2 Fig). These results showed that remimazolam-induced calcium elevation was not induced by morphological changes in the ER of both SHSY-5Y cells and HUVECs, unlike our previous research using propofol.

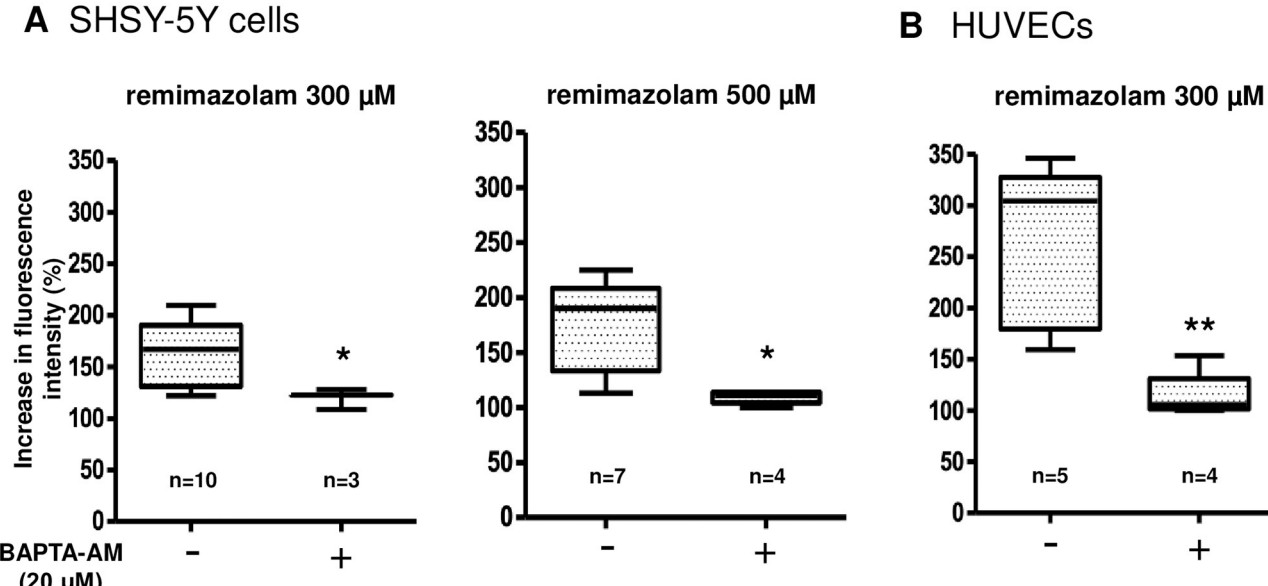

**Fig 4. Characterization of remimazolam-induced calcium elevation in the absence of intracellular calcium.** (A) BAPTA-AM (20 μM), a calcium chelator, significantly reduced and almost abolished the 300 and 500 μM remimazolam-induced calcium elevation in SHSY-5Y cells (n = 3–10, * p = 0.028, compared to control, Mann Whitney test, and n = 4–7, * p = 0.0242, compared to control, Mann Whitney test, respectively). The horizontal line in each box indicates the median, the box shows the interquartile range (IQR), and the whiskers represent 1.5 × IQR. (B) BAPTA-AM (20 μM), a calcium chelator, significantly reduced and almost abolished the remimazolam (300 μM)-induced calcium elevation in HUVECs (n = 4–5, * p = 0.0159, compared to control, Mann Whitney test). The horizontal line in each box indicates the median, the box shows the interquartile range (IQR), and the whiskers represent 1.5 × IQR.

### 3.6. Mechanism underlying remimazolam-induced mobilization of calcium from the ER

**3.6.1. Possible involvement of GPCRs in remimazolam-induced calcium elevation.** Next, we investigated the mechanisms underlying remimazolam-induced intracellular calcium elevation. To elucidate the possible involvement of G-protein coupled receptors (GPCRs), we examined the effects of U-73122, a phospholipase C (PLC) inhibitor, on remimazolam-induced calcium elevation. Pretreatment with U-73122 at 5 μM significantly inhibited the 10 μM acetylcholine-induced calcium elevation through muscarinic receptors expressed in SHSY-5Y cells (Fig 6A) as well as the 10 μM histamine-induced calcium elevation through histamine (H1) receptors expressed in HUVECs (Fig 6B).

Similarly, U-73122 significantly inhibited remimazolam-induced calcium elevation in both SHSY-5Y cells and HUVECs (Fig 6A and 6B). These results suggest that the elevation of intracellular calcium by remimazolam may be mediated by GPCRs coupled with PLC.

**3.6.2. Possible involvement of IP$_3$ and ryanodine receptors in remimazolam-induced calcium elevation.** Calcium mobilization from the ER is mediated by inositol 1,4,5-triphosphate (IP$_3$) or ryanodine receptors (RyRs). Therefore, we examined the effects of xestospongin C (Xc), an IP$_3$ receptor (IP$_3$R) antagonist, on remimazolam-induced calcium elevation. Pretreatment with Xc at 5 μM significantly inhibited the 10 μM acetylcholine-induced calcium elevation in SHSY-5Y cells (Fig 7A) and 10 μM histamine-induced calcium elevation in HUVECs (Fig 7B).

Similarly, 5 μM Xc significantly reduced the remimazolam-induced calcium elevation in both SHSY-5Y cells and HUVECs (Fig 7A and 7B). These results suggest that the elevation of intracellular calcium by remimazolam may be mediated by IP$_3$ receptors.

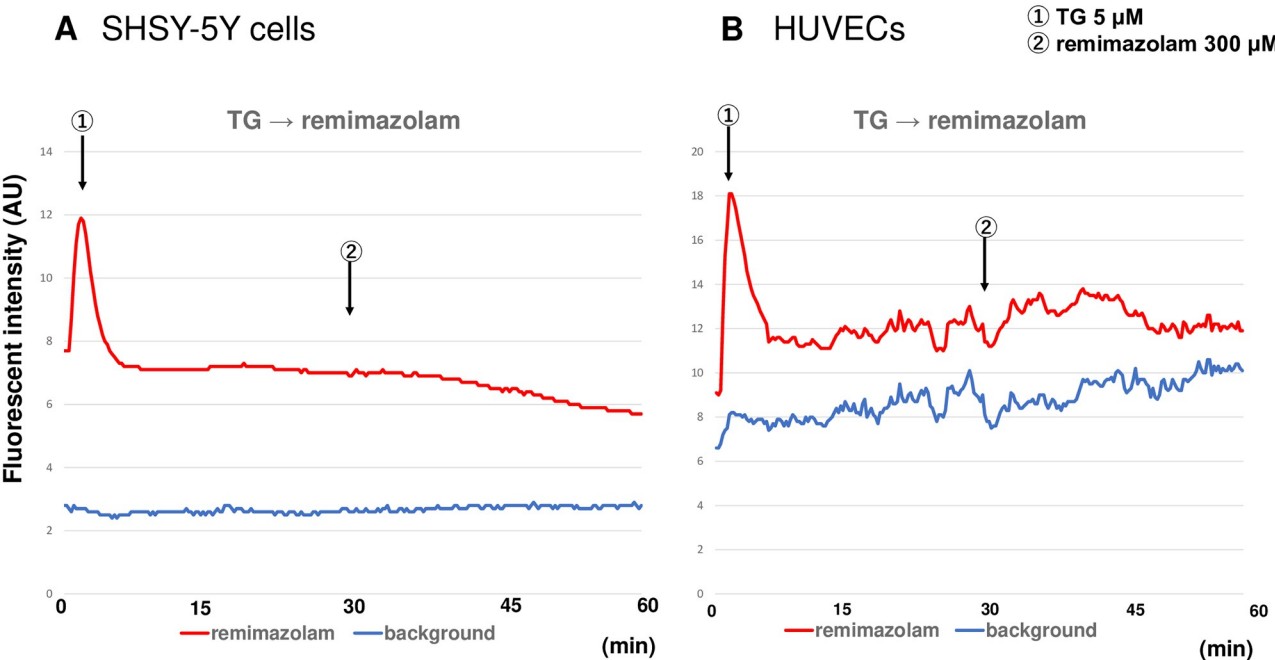

**Fig 5. Effects of thapsigargin on remimazolam-induced calcium elevation.** (A) Following the administration of 5 μM thapsigargin (TG), a $Ca^{2+}$-ATPase inhibitor (①), the $[Ca^{2+}]_i$ was elevated in SHSY-5Y cells. Subsequently, remimazolam at 300 μM was administered 30 min after the TG administration (②). Remimazolam did not induce any calcium elevation, suggesting that remimazolam mobilized calcium from ER. (B) Following the administration of 5 μM thapsigargin (TG), a $Ca^{2+}$-ATPase inhibitor (①), the $[Ca^{2+}]_i$ was elevated in HUVECs. Subsequently, remimazolam at 300 μM was administered 30 minutes after the TG administration (②). Remimazolam did not induce any calcium elevation, suggesting that remimazolam mobilized calcium from ER. For both experiments, data from one representative experiment are shown.

In addition, we investigated the effects of dantrolene, a selective RyR antagonist, on remimazolam-induced calcium elevation. First, we examined dantrolene effects on calcium mobilization induced by caffeine, a RyR agonist. The results showed that pretreatment with 50 μM dantrolene significantly inhibited 10 mM caffeine-induced calcium mobilization in both SHSY-5Y cells and HUVECs (Fig 8A and 8B).

However, dantrolene did not significantly influence remimazolam-induced calcium elevation in either cell lines (Fig 8A and 8B), suggesting that remimazolam-induced calcium elevation may not be mediated *via* RyRs. We also examined the effects of tetracaine, another type of RyR antagonist, on remimazolam-induced calcium elevation in SHSY-5Y cells. Tetracaine (500 μM) did not significantly affect the calcium elevation, although it significantly inhibited caffeine (10 mM)-induced calcium elevation (S3 Fig).

## 4. Discussion

Remimazolam has already been used in many clinical settings since it was approved in Japan in July 2020, prior to the rest of the world. Its structure is comparable to that of the existing benzodiazepines. Its sedative effect is on GABA_A receptors, and it has relatively little effect on circulation. One of the benefits of remimazolam is that its actions can be antagonized by flumazenil [1, 2]. At this point, remimazolam is only approved for Induction and maintenance of general anesthesia, but it is likely that it will play a leading role in intravenous anesthetics in the future, similar to propofol.

Remimazolam and propofol are considered to exert their anesthetic effects by acting on GABA_A receptors. It has been suggested that elevated intracellular calcium is involved in the

**A**  SHSY-5Y cells

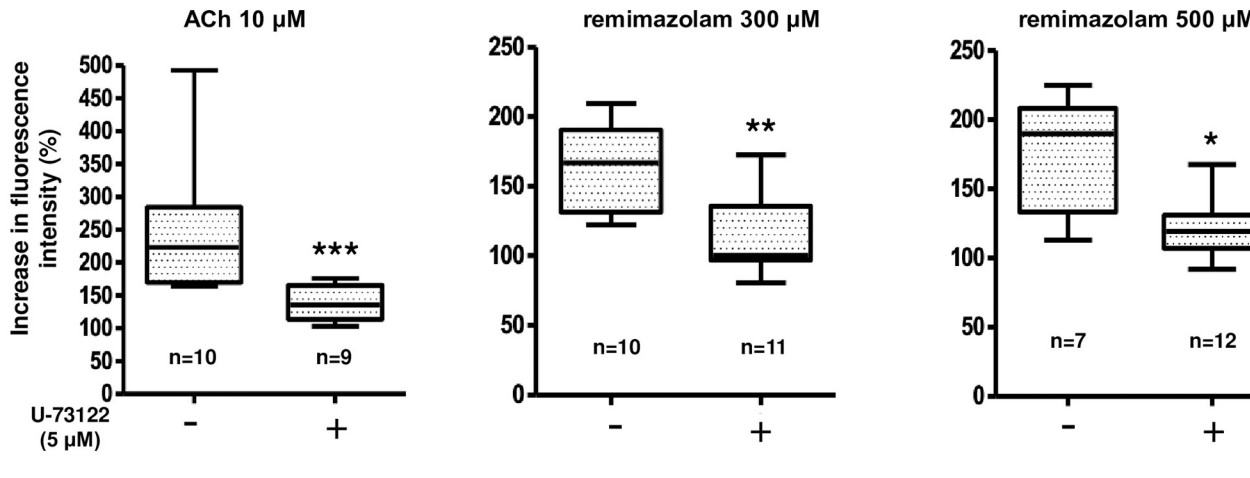

**B**  HUVECs

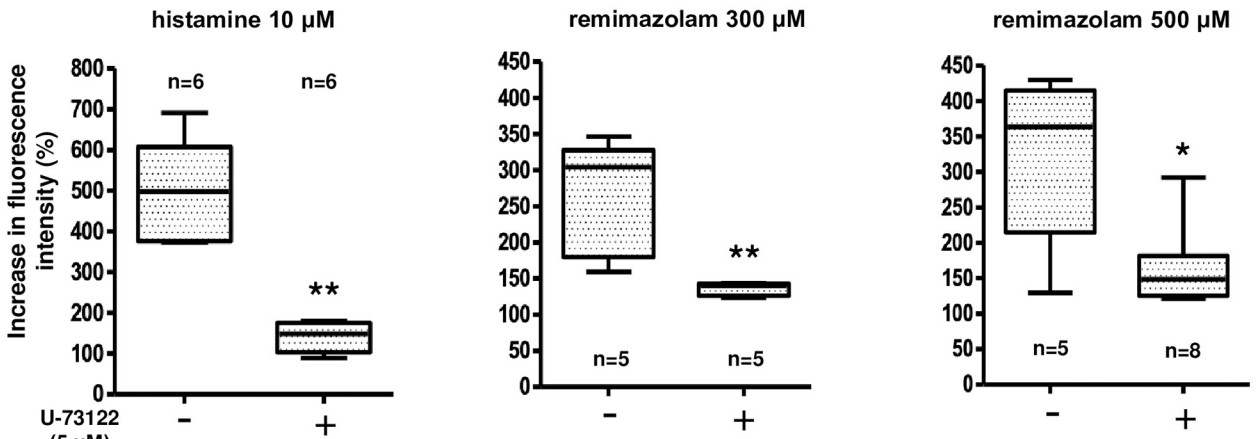

**Fig 6. Characterization of remimazolam-induced calcium elevation; possible involvement of the G-protein coupled receptors.** (A) Fifteen minutes-pretreatment with U-73122, a phospholipase C (PLC) inhibitor, at 5 μM significantly inhibited the 10 μM acetylcholine (ACh)-induced calcium elevation in SHSY-5Y cells (n = 9–10, *** p = 0.0004, compared to control, Mann Whitney test). Similarly, 15 minutes-pretreatment with U-73122 significantly affect 300 and 500 μM remimazolam-induced calcium elevation in SHSY-5Y cells (n = 9–10, ** p = 0.0044, compared to control, Mann Whitney test, and n = 7–12, * p = 0.018, compared to control, Mann Whitney test, respectively). The horizontal line in each box indicates the median, the box shows the interquartile range (IQR), and the whiskers represent 1.5 × IQR. (B) Fifteen minutes-pretreatment with U-73122 at 5 μM significantly inhibited the 10 μM histamine-induced calcium elevation in HUVECs (n = 6, ** p = 0.0022, compared to control, Mann Whitney test). Similarly, 15 minutes-pretreatment with U-73122 significantly affected 300 and 500 μM remimazolam-induced calcium elevation in HUVECs (n = 5, ** p = 0.0079, compared to control, Mann Whitney test, and n = 5–8, * p = 0.0295, compared to control, Mann Whitney test, respectively). The horizontal line in each box indicates the median, the box shows the interquartile range (IQR), and the whiskers represent 1.5 × IQR.

development of adverse effects of propofol. We have previously reported that propofol mobilizes calcium into cells by penetrating them and causing morphological changes in intracellular organelles [6]. In our recent report, we also showed that high concentrations of remimazolam increase intracellular calcium concentration in a dose-dependent manner [7]. In this study, we

**A** SHSY-5Y cells

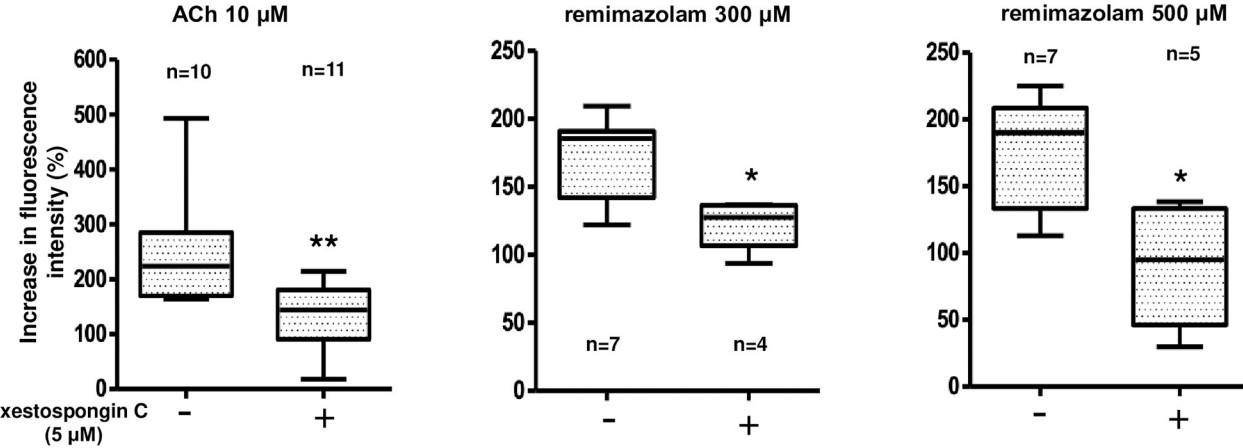

**B** HUVECs

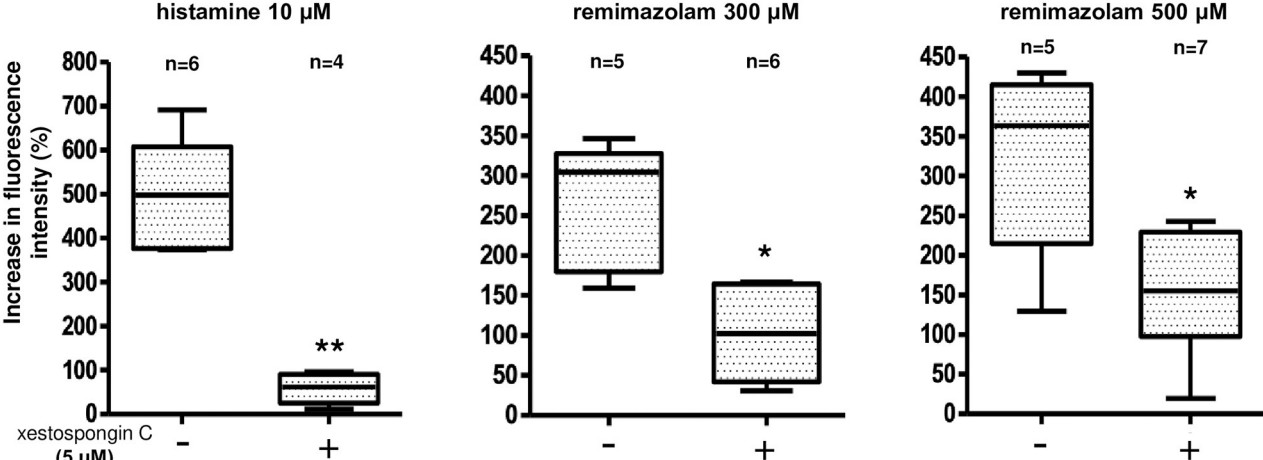

**Fig 7. Characterization of remimazolam-induced calcium elevation; possible involvement of the IP₃ receptors.** (A) Fifteen minutes-pretreatment with Xestospongin C (Xc), an inositol 1,4,5-triphosphate receptor (IP₃R) antagonist, at 5 µM significantly inhibited the ACh 10 µM-induced calcium elevation in SHSY-5Y cells (n = 10–11, ** p = 0.0035, compared to control, Mann Whitney test). Similarly, 15 minutes-pretreatment with Xc at 5 µM affected 300 and 500 µM remimazolam-induced calcium elevation in SHSY-5Y cells (n = 4–7, * p = 0.0242, compared to control, Mann Whitney test, and n = 5–7, * p = 0.0177, compared to control, Mann Whitney test, respectively). The horizontal line in each box indicates the median, the box shows the interquartile range (IQR), and the whiskers represent 1.5 × IQR. (B) Fifteen minutes-pretreatment with Xestospongin C (Xc) at 5 µM significantly inhibited the histamine 10 µM-induced calcium elevation in HUVECs (n = 4–6, ** p = 0.0095, compared to control, Mann Whitney test). Similarly, 15 minutes-pretreatment with Xc at 5 µM affected 300 and 500 µM remimazolam-induced calcium elevation in HUVECs (n = 5–6, * p = 0.0173, compared to control, Mann Whitney test, and n = 5–7, * p = 0.0303, compared to control, Mann Whitney test, respectively). The horizontal line in each box indicates the median, the box shows the interquartile range (IQR), and the whiskers represent 1.5 × IQR.

examined the effects of remimazolam on the dynamics of intracellular calcium in each cell, including the morphology of intracellular organelles.

In the present work, we used SHSY-5Y cells and HUVECs derived from neuronal and endothelial cells, respectively, which are considered important sites of action for intravenous

**A** SHSY-5Y cells

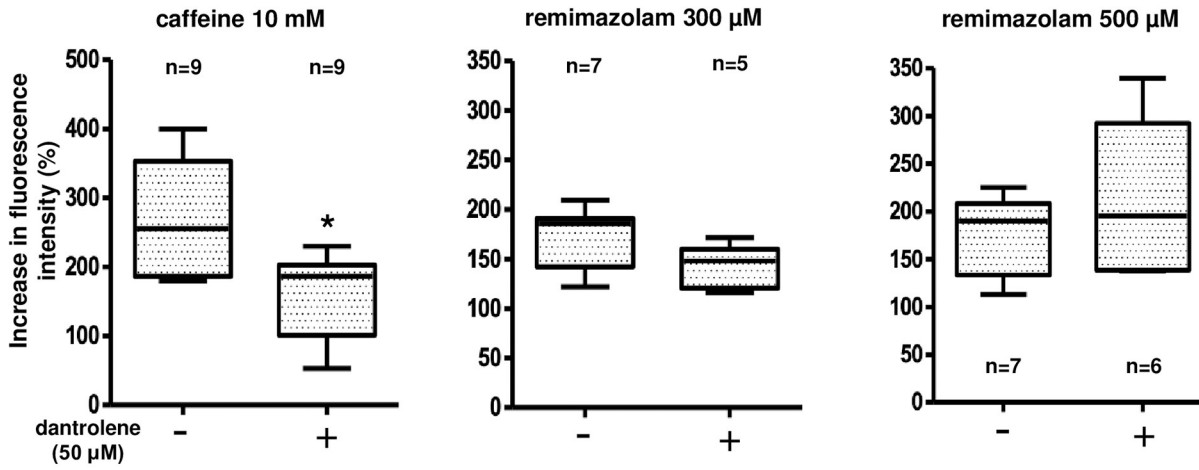

**B** HUVECs

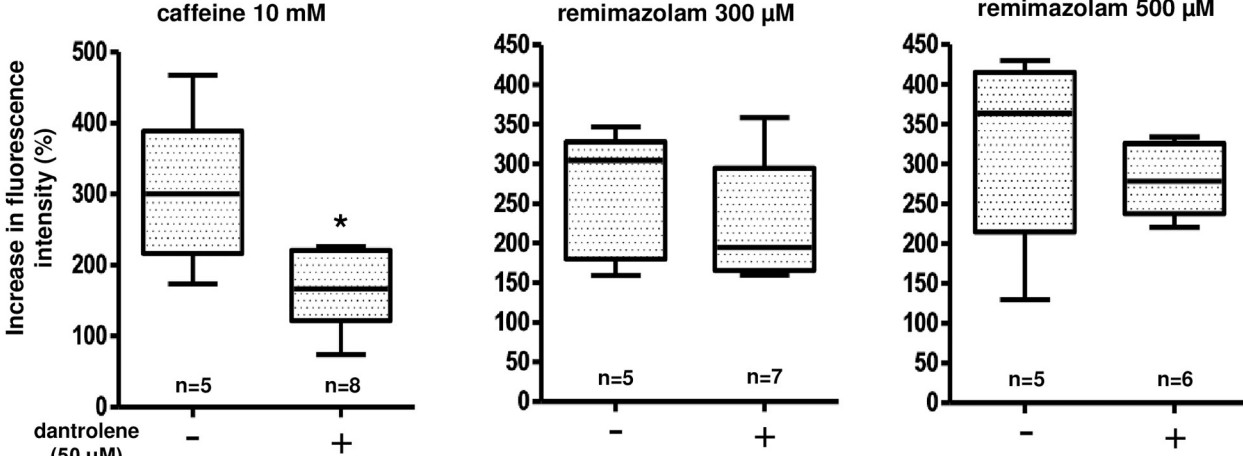

**Fig 8. Characterization of remimazolam-induced calcium elevation; possible involvement of ryanodine receptors.** (A) Caffeine, a ryanodine receptor (RYR) agonist, at 10 mM induced the intracellular calcium in SHSY-5Y cells. Fifteen minutes-pretreatment with dantrolene, a RYR antagonist, at 50 μM significantly inhibited the caffeine-induced calcium mobilization (n = 9, * p = 0.024, compared to control, Mann Whitney test), however, 15 minutes-pretreatment with dantrolene at 50 μM did not significantly influence the 300 and 500 μM remimazolam-induced calcium elevation (n = 5–7, p = 0.149, compared to control, Mann Whitney test, and n = 6–7, p = 0.5338, compared to control, Mann Whitney test, respectively). The horizontal line in each box indicates the median, the box shows the interquartile range (IQR), and the whiskers represent 1.5 × IQR. (B) Caffeine at 10 mM induced the intracellular calcium in HUVECs. Fifteen minutes-pretreatment with dantrolene at 50 μM significantly inhibited the caffeine-induced calcium mobilization (n = 5–8, * p = 0.0186, compared to control, Mann Whitney test), however, 15 minutes-pretreatment with dantrolene at 50 μM did not significantly influence the 300 and 500 μM remimazolam-induced calcium elevation (n = 5–7, p = 0.5303, compared to control, Mann Whitney test, and n = 5–7, p = 0.2468, compared to control, Mann Whitney test, respectively). The horizontal line in each box indicates the median, the box shows the interquartile range (IQR), and the whiskers represent 1.5 × IQR.

anesthetics. Although the therapeutic plasma concentrations are considerably lower than those used in our experiments, the results showed that remimazolam increased intracellular calcium in a dose-dependent manner in both cell lines. These results were similar to those of our previous experiments in which propofol was used [6]. In addition to SHSY-5Y cells and HUVECs, remimazolam-induced intracellular calcium elevation was observed in HEK293, suggesting that this effect is universal, regardless of the cell type.

Next, we examined how remimazolam-induced intracellular calcium elevation could occur. When remimazolam was administered using extracellular calcium-free buffer, we observed an increase in intracellular calcium levels in both SHSY-5Y cells and HUVECs. Therefore, it is unlikely that this increase was caused by a calcium influx from extracellular sources.

We further confirmed that the remimazolam-induced elevation of intracellular calcium could be largely eliminated by BAPTA-AM in both SHSY-5Y cells and HUVECs. Moreover, the remimazolam-induced increase in intracellular calcium was almost eliminated by removing calcium from the ER by thapsigargin. Based on these findings, we hypothesized that the remimazolam-induced increase in intracellular calcium levels might be due to calcium mobilization from the ER. We subsequently attempted to elucidate the mechanism of calcium mobilization from the ER. Since GPCR-IP$_3$ and ryanodine receptors may be involved in the mechanism of calcium mobilization from the ER, we investigated the effects of a PLC inhibitor (U-73122), an IP$_3$ receptor inhibitor (Xc), and a ryanodine receptor inhibitor (dantrolene). U-73122 and Xc significantly inhibited remimazolam-induced intracellular calcium elevation in both cell lines, whereas dantrolene had no effect, suggesting that this elevation is involved in the GPCR-IP$_3$ pathway. While remimazolam does not seems to affect RyR1 directly, the IP$_3$-dependent calcium elevation mediated by remimazolam could play a role in sensitizing RyR1 in an individual with Malignant Hyperthermia (MH) susceptibility. A recent study reported that there were no changes in the sensitivity to intracellular calcium elevation of remimazolam in cells transfected with the MH mutation [7]. This is very different from propofol-induced intracellular calcium elevation, the mechanism of which is involved in the direct destruction of intracellular organelles, such as the ER. In this meaning, the effects of remimazolam might be reversible since it elevates calcium by pharmacological mechanisms rather than by non-reversible morphological changes.

In this study, we could not identify the type of GPCRs affected. Remimazolam-induced elevation of intracellular calcium was observed in all the investigated cells, including SHSY5Y, HUVECs, and HEK293 (S2 Fig). Based on these results, it is expected that GPCRs, which are related to remimazolam action, are universally expressed in a variety of cells. As an example, we considered the involvement of purinergic receptors, particularly P2 purinergic receptor [9]. In the future, it is expected that studies will be performed to identify GPCRs involved in remimazolam-induced calcium elevation by focusing on GPCRs universally expressed by these cells.

Remimazolam concentrations used in this study were considerably higher than those used clinically (appropriately 5 μM for maintaining anesthesia) [10]. However, concentrations like the ones used in this study or higher could occur locally near the injection site. Currently, the indication for remimazolam is limited to general anesthesia in adults, but it is expected that this indication will be expanded to general anesthesia for pediatric patients in the future. Since the early 2000s, many basic studies on neurological disorders caused by general anesthetics have been conducted, and it has been reported that many general anesthetics have neurotoxic effects on juvenile central neurons and are associated with occurrence of developmental disorders in the long-term [11–14]. In such cases, prolonged exposure can lead to the accumulation of high concentrations of anesthetics in the cells, causing side effects. It is well-known that GABA$_A$ receptors, NMDA receptors, and IP$_3$ receptors are associated with the receptors to

which general anesthetics bind [15, 16]. Evidence has shown that overexposure or prolonged exposure of general anesthetics to these receptors induces an excessive increase in intracellular calcium and consequently, apoptosis [17–20]. Therefore, it is worthwhile to investigate the effects of high concentrations of remimazolam on cellular functions. Our findings that high concentrations of remimazolam universally elevated intracellular calcium levels will benefit future research.

## 5. Conclusion

Our present data indicate that high concentrations of remimazolam induced a dose-dependent elevation of intracellular calcium in SHSY-5Y cells and HUVECs. We also found that remimazolam acted on the GPCR-IP$_3$ pathway to increase intracellular calcium levels. As the clinical uses of remimazolam are expected to expand worldwide in the future, elucidating its mechanism of action is very important to understand its advantages and disadvantages.

## Supporting information

**S1 Fig. Remimazolam-induced intracellular calcium elevation in HEK293 cells.** (A) Remimazolam at a concentration greater than or equal to 300 μM significantly induced the elevation of intracellular calcium in a dose-dependent manner in HEK293 cells. (n = 3–5, * p < 0.05, compared to control, one-way ANOVA followed by Dunnett's post-test).
(TIF)

**S2 Fig. Fluorescent microscopic imaging.** ER-tracker was capable of staining ER in living HUVECs and SHSY-5Y cells, remimazolam did not elicit the morphological changes of ER. (above: HUVECs, below: SHSY-5Y cells).
(TIF)

**S3 Fig. Characterization of remimazolam-induced calcium elevation; possible involvement of ryanodine receptors.** Caffeine, a ryanodine receptor (RYR) agonist, at 10 mM induced the intracellular calcium in SHSY-5Y cells. Fifteen minutes-pretreatment with tetracaine, a RYR antagonist, at 500 μM significantly inhibited the caffeine-induced calcium mobilization (n = 5, * p = 0.0159, compared to control, Mann Whitney test), however, 15 minutes-pretreatment with tetracaine at 500 μM did not significantly influence the 300 μM remimazolam-induced calcium elevation. Data represent the mean ± SEM (n = 4, p = 0.3429, compared to control, Mann Whitney test).
(TIF)

## Acknowledgments

We thank Mundipharma Japan and PAION Deutschland GmbH for the generous gift of remimazolam. This work was performed using equipment at the Radiation Research Center for Frontier Science, Natural Science Center for Basic Research and Development, Hiroshima University, and Biosignal Research Center, Kobe University, Japan. All authors approved the final version of the manuscript for publication.

## Author Contributions

**Conceptualization:** Tomoaki Urabe, Hirotsugu Miyoshi.

**Data curation:** Tomoaki Urabe, Hirotsugu Miyoshi, Norio Sakai.

**Investigation:** Tomoaki Urabe, Soshi Narasaki, Norio Sakai.

**Resources:** Soshi Narasaki, Yuhki Yanase, Kazue Uchida, Soma Noguchi, Norio Sakai.

**Supervision:** Hirotsugu Miyoshi, Michihiro Hide, Yasuo M. Tsutsumi, Norio Sakai.

**Writing – original draft:** Tomoaki Urabe.

**Writing – review & editing:** Norio Sakai.

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
