## [Decision Letter · Decision Letter 0]

14 Sep 2021

PONE-D-21-15802Characterization of intracellular calcium mobilization induced by remimazolam, a newly approved intravenous anestheticPLOS ONE

Dear Dr. Miyoshi,

Thank you for submitting your manuscript to PLOS ONE. After careful consideration, we feel that it has merit but does not fully meet PLOS ONE’s publication criteria as it currently stands. Therefore, we invite you to submit a revised version of the manuscript that addresses the points raised during the review process.

In Your revision please pay special attention to the criticism formulated by Reviewer#1 regarding the detection of intracellular calcium concentration changes using a non-ratiometric dye. Please re-evaluate the data based on those comments, especially how the fluorescent values are calculated (background correction, use of F/F0, etc.).

We look forward to receiving your revised manuscript.

Kind regards,

Laszlo Csernoch, Ph.D.

Academic Editor

PLOS ONE

Journal Requirements:

2. Please provide additional information about the source of the HUVEC cells. If the cells are a named cell line, please provide the name. If they were primary cells derived from clinical samples, please state whether the cells were isolated for the purposes of research and whether any identifying information was provided with them.

5. PLOS requires an ORCID iD for the corresponding author in Editorial Manager on papers submitted after December 6th, 2016. Please ensure that you have an ORCID iD and that it is validated in Editorial Manager. To do this, go to ‘Update my Information’ (in the upper left-hand corner of the main menu), and click on the Fetch/Validate link next to the ORCID field. This will take you to the ORCID site and allow you to create a new iD or authenticate a pre-existing iD in Editorial Manager. Please see the following video for instructions on linking an ORCID iD to your Editorial Manager account: https://www.youtube.com/watch?v=_xcclfuvtxQ.

7. We noticed you have some minor occurrence of overlapping text with the following previous publication(s), which needs to be addressed:

-https://www.sciencedirect.com/science/article/abs/pii/S0014299920303952?via%3Dihub

In your revision ensure you cite all your sources (including your own works), and quote or rephrase any duplicated text outside the methods section. Further consideration is dependent on these concerns being addressed.

Reviewers' comments:

Reviewer's Responses to Questions

**Comments to the Author**

1. Is the manuscript technically sound, and do the data support the conclusions?

Reviewer #1: Partly

Reviewer #2: Partly

2. Has the statistical analysis been performed appropriately and rigorously? 

Reviewer #1: No

Reviewer #2: No

3. Have the authors made all data underlying the findings in their manuscript fully available?

Reviewer #1: No

Reviewer #2: No

4. Is the manuscript presented in an intelligible fashion and written in standard English?

Reviewer #1: Yes

Reviewer #2: Yes

5. Review Comments to the Author

Reviewer #1: This manuscript aimed to examine the source of increased Ca2+ in response to exposure to the drug remimazolam. The authors have done a lot of work and have endeavoured to pharmacologically block multiple cellular pathways that could account for the Ca2+ release observed. Ultimately, the authors suggest that the increased Ca2+ is likely mediated via a GPCR-IP3 pathway, although more work needs to be done to establish the specific pathway.

Major Points:

• What is the rational for the choices of cell lines used? One of the cell lines reported, but no data was shown, is from a monkey kidney line.

• The introduction is quite brief with minimal references provided throughout the submission.

• A high proportion of supportive data is not shown as figures. For example, it is reported as data not shown but how did the authors designate a “normal morphology” of the ER in the SHSY-5Y and HUVEC cells? Images would help substantiate this statement.

• What is the equivalent clinical dose for remimazolam? The authors state that the doses they use are higher than those used in clinical situations but seem to only get cellular effects at with high doses. How do these experimental doses correspond to the maximum dosage a patient would receive?

• A non-ratiometric, AM loading, Ca2+ indicating dye has been used. What is the justification for the ratio that was created for the Ca2+ trace analysis? Convention is to subtract the background intensity and then convert to F/F0 to take into account the differences in dye loading between individual cells and different cell lines. The summarised percentage change in fluorescence may not reflect a large change in Ca2+ concentration due to the use of a non-ratiometric dye. It is also not appropriate to apply statistics to percentage data.

• 30 mins after the TG treatment an agonist should be applied to show that there is no residual Ca2+ in the stores to further support the assumption that remimazolam is not causing a release of ER Ca2+.

• The authors state that Ca2+ release due to remimazolam is likely due to IP3 receptor activation. It would be beneficial to apply a cocktail of the inhibitors used in this study (minus the IP3 inhibitor) and see if the same increased Ca2+ response is seen with treatment with remimazolam.

• In Figure2 and 3, can the authors comment on why the same trends of Ca2+ release were not observed in the HUVEC cells but are conserved in the SHSY-5Y?

• Figure 8 A&B the data suggests that dantrolene was not providing a complete block of the RyR. Tetracaine would be a more efficient inhibitor.

Minor Points:

• Histogram formats should be changed to include individual data points.

• Figure 1, states “the rate” of calcium, is this a typo that should state ratio as the authors have not measured the rate of Ca2+ release/increase?

• Figure 4, it appears the data panel of HUVECS with 500uM remimazolam is absent, was the data unable to be gathered under these conditions?

Reviewer #2: Urabe et al. provide important information about the effect of remimazolam, a newly approved intravenous anesthetic. This work presents an empirical characterization of intracellular calcium alterations initiated by this new drug. The authors present results on calcium signaling using different mammalian cell lines such as neuronal-, fibroblast-like cells and human umbilical endothelial cells. Using calcium imaging, ionic substitution, calcium buffering, and pharmacological tools, the authors identified that supratherapeutic doses of remimazolam initiated intracellular calcium elevations from the ER. These Ca signals were GPCR and IP3 dependent when assessed in several cell types. The work is well-designed and presents empirical evidence of an interesting biological event that has a therapeutic interest.

For this reviewer, the work is novel and original, necessary for the general understanding of the additional (side) effects of remimazolam. The experiments are, in general, well-conducted and the paper is well written and easy to follow; however, as far as this reviewer concerns, some issues require attention.

*Specific Comments

Abstract:

Briefly mention that remimazolam causes an elevation in intracellular calcium and why this may be of interest.

Introduction:

Line 75. Propose a mechanism by which propofol (or remimazolam) could cause hypotension or vascular pain. This mechanism could be used to justify the study of calcium mobilization and the use of neuronal-like cells and endothelial cells.

Methods:

Lines 126-130. For reproducibility purposes, please provide more details about the apparatus and software used to measure calcium signals. Was a photodiode, PMT, or camera used (include manufacturer and model)? Also, add details about excitation light source, excitation, and emission filters, and objective used. What was the sampling rate?

Statistical: Please clarify whether the normality test was assessed and how. When reporting results (significant or non-significant), please give the exact p-value, adjusted significance level (when appropriate), and the test used.

Results:

Lines 194-204. “Observation of remimazolam…” Delete this section or show the results, a negative results still a result. Data could be show as supplementary information.

To further characterize the involvement of GPCR, the authors could use GDPbetaS, to prevent non-specific activation of GPCR. If remimazolam effect depend on GPCR activation GDPbetaS should prevent its effects.

Discussion:

Lines 262. “ a newly marketed” delete this, it has been already mention few lines above.

The authors wrote: “Remimazolam concentrations used in this study were considerably higher than those used clinically.”

You could add something like: However, concentrations like the ones used in this study or higher could occur locally near the injection site.

It will be worth mentioning somewhere in the discussion that while remimazolam does not seems to affect RyR1 directly, the IP3-dependent elevation in Ca2+ mediated by remimazloam could play a role in sensitizing RyR1 in an individual with HM-susceptibility.

Figure 1:

Traces in panel B appear identical to those previously published by the authors (see Ref #6 Urabe T et al. 2020, Fig. 1B). This issue is very problematic and needs to be fixed or request permission to reuse already published data. Also, in the same panel, how comes that the background signal is not flat. Would you please indicate the region of interest used to measure both the cell and background on the image?

Figures 2, 3, 4, 6-8.

Use box plot with data overlap instead of plunger bars (show the data).

Figures 2 and 3 add the number of observations to boxes as in figures 4,6-8.

Minor Comments:

Figure 2-8. Adding labels to the figure panels (i.e., Control solution for Fig 2, Zero Ca2+ for figure 3) may help the reader quickly identify the results.

Figure 3A, remove data for DMSO 0.05-0.3 as in panel B.

Figure 4. Increase Font size for BAPTA-AM and symbols

Figures 6-8. Increase the font size for x- and y-axis labels

6. PLOS authors have the option to publish the peer review history of their article (what does this mean?). If published, this will include your full peer review and any attached files.

Reviewer #1: No

Reviewer #2: No

---

## [Author Response · Author response to Decision Letter 0]

1 Dec 2021

GENERAL COMMENTS TO THE EDITOR

COMMENTS TO THE EDITOR:

1. COMMENT: Please ensure that your manuscript meets PLOS ONE's style requirements, including those for file naming.

RESPONSE: I have checked and corrected. Fig legend has been added after each Fig.

2. COMMENT: Please provide additional information about the source of the HUVEC cells. If the cells are a named cell line, please provide the name. If they were primary cells derived from clinical samples, please state whether the cells were isolated for the purposes of research and whether any identifying information was provided with them.

RESPONSE: We have added information on the availability of HUVECs. （page 7, line 107 - 108）

3. COMMENT: We note that the grant information you provided in the ‘Funding Information’ and ‘Financial Disclosure’ sections do not match. When you resubmit, please ensure that you provide the correct grant numbers for the awards you received for your study in the ‘Funding Information’ section.

RESPONSE: We have added our Funding Information including the grant numbers in our cover letter. 

4. COMMENT: In your Data Availability statement, you have not specified where the minimal data set underlying the results described in your manuscript can be found. PLOS defines a study's minimal data set as the underlying data used to reach the conclusions drawn in the manuscript and any additional data required to replicate the reported study findings in their entirety. All PLOS journals require that the minimal data set be made fully available. 

RESPONSE: We have attached the data as Supporting information.

5. COMMENT: PLOS requires an ORCID iD for the corresponding author in Editorial Manager on papers submitted after December 6th, 2016. Please ensure that you have an ORCID iD and that it is validated in Editorial Manager. To do this, go to ‘Update my Information’ (in the upper left-hand corner of the main menu), and click on the Fetch/Validate link next to the ORCID field. This will take you to the ORCID site and allow you to create a new iD or authenticate a pre-existing iD in Editorial Manager.

RESPONSE: Corresponding author obtained the ORCID ID and confirmed that it is enabled in Editorial Manager.

6. COMMENT: We note that you have included the phrase “data not shown” in your manuscript. Unfortunately, this does not meet our data sharing requirements. PLOS does not permit references to inaccessible data. We require that authors provide all relevant data within the paper, Supporting Information files, or in an acceptable, public repository. Please add a citation to support this phrase or upload the data that corresponds with these findings to a stable repository (such as Figshare or Dryad) and provide and URLs, DOIs, or accession numbers that may be used to access these data. Or, if the data are not a core part of the research being presented in your study, we ask that you remove the phrase that refers to these data.

RESPONSE: We have presented the data as Supporting information. The description of the paper has been corrected accordingly. (page 12, line 199; page 16, line 289; page 20, line 391; page 24, line 453)

7. COMMENT: We noticed you have some minor occurrence of overlapping text with the following previous publication(s), which needs to be addressed:

https://www.sciencedirect.com/science/article/abs/pii/S0014299920303952?via%3Dihub

RESPONSE: We have identified that many of the duplications overlap with papers we have published in the past. We have made every effort to minimize duplication where possible.

 

GENERAL COMMENTS TO THE REVIEWER #1

RESPONSES TO THE REVIEWER #1’s COMMENTS:

Major points

1. COMMENT: What is the rational for the choices of cell lines used? One of the cell lines reported, but no data was shown, is from a monkey kidney line.

RESPONSE: SHSY-5Y cells are neuronal cell lines; HUVECs are vascular endothelial cells. Remimazolam is expected to have anesthetic effects and side effects on neuronal and vascular endothelial cells. Hence, we used these cells in our study. The other cells were used to determine if the effects of remimazolam are specific to neurons and endothelial cells or if it is a generalized effect. There was not enough data accumulated for COS-7 and Hela cells to provide examples, so the descriptions of these cells were removed in the revised paper.

2. The introduction is quite brief with minimal references provided throughout the submission.

RESPONSE: In accordance with the reviewer's comment, we added a new description of the side effects of anesthetics in the introduction.

3. COMMENT: A high proportion of supportive data is not shown as figures. For example, it is reported as data not shown but how did the authors designate a “normal morphology” of the ER in the SHSY-5Y and HUVEC cells? Images would help substantiate this statement.

RESPONSE: We have presented data showing that administration of remimazolam did not cause morphological changes in the ER as Supplemental figure 1. (page 15 - 16, line 286 - 289)

4. COMMENT: What is the equivalent clinical dose for remimazolam? The authors state that the doses they use are higher than those used in clinical situations but seem to only get cellular effects at with high doses. How do these experimental doses correspond to the maximum dosage a patient would receive?

RESPONSE: Although the concentration of remimazolam used in this study is certainly higher than that of clinical concentration, there is a high possibility that remimazolam will be expanded to pediatric general anesthesia in the future. In pediatric patients, various neurological disorders are likely to occur during long-term exposure to anesthetics. Given the possibility that remimazolam will be accumulated due to long-term exposure, remimazolam may exert its effects at higher than clinical concentrations. Therefore, the findings of our experiments may have clinical significance. These ideas have already been mentioned in Discussion.

5. COMMENT: A non-ratiometric, AM loading, Ca2+ indicating dye has been used. What is the justification for the ratio that was created for the Ca2+ trace analysis? Convention is to subtract the background intensity and then convert to F/F0 to take into account the differences in dye loading between individual cells and different cell lines. The summarised percentage change in fluorescence may not reflect a large change in Ca2+ concentration due to the use of a non-ratiometric dye. It is also not appropriate to apply statistics to percentage data.

RESPONSE: We do not intend to measure the absolute intracellular calcium concentration. We are using this method only as a semi-quantification method that can be used for the purpose of analyzing the effects of drugs on intracellular calcium elevation. As reported in our previous research paper using propofol (Urabe T, et al. Eur J Pharmacol. 2020;884:173303.), time-lapse imaging was carried out using a fluorescence microscope after the application of propofol in order to observe propofol-triggered calcium elevation.

In this previous study, to examine the validity of this method, we have confirmed that the number of cells in the observation area does not influence the degree of drug-induced intracellular calcium elevation. Therefore, we believe that this semi-quantification method can be used to determine the effects of various drugs on the calcium elevation of remimazolam. Since we averaged the calcium elevation of all cells in an area, we believe this method would be less biased and more accurate, compared to averaging the calcium elevations of intentionally picked individual cells, in measuring the degree of calcium elevation.

In this study, we showed the percentage increase in the fluorescence intensity of the entire area by remimazolam before administration of remimazolam. We have no intention to measure absolute values of fluorescence intensity. We performed a number of experiments with different experimental conditions, calculated the percentage increase in presentation for each experiment, and compared them across experiments. This type of analysis is general, not limited to calcium elevation experiments, and we do not believe that the use of statistical methods is inappropriate. We have shown the area where the background was measured in Fig. 1 in order to give an accurate understanding of the method used in this experiment.

6. COMMENT: 30 mins after the TG treatment an agonist should be applied to show that there is no residual Ca2+ in the stores to further support the assumption that remimazolam is not causing a release of ER Ca2+.

 RESPONSE: In our previous study, we found that propofol mobilized intracellular calcium from ER by disrupting the membrane of the endoplasmic reticulum (Urabe T, et al. Eur J Pharmacol. 2020;884:17330). In this previous study, we investigated the effect of propofol on the mobilization of calcium from ER treated with TG. We proved that propofol did not increase intracellular calcium 30 minutes after the TG administration, indicating that 30-min treatment with TG sufficiently depleted calcium in ER. Therefore, we believe that calcium in the endoplasmic reticulum is thoroughly depleted at 30 minutes after TG administration.

7. COMMENT: The authors state that Ca2+ release due to remimazolam is likely due to IP3 receptor activation. It would be beneficial to apply a cocktail of the inhibitors used in this study (minus the IP3 inhibitor) and see if the same increased Ca2+ response is seen with treatment with remimazolam.

RESPONSE: The reviewer has proposed to use a cocktail containing an inhibitor other than the IP3 inhibitor used in this study, and to test whether intracellular calcium is elevated in the presence of this inhibitor cocktail. However, since the cocktail contains PLC inhibitors, the results would probably show that no calcium increase will be observed. It is unlikely that the results of this experiment will provide beneficial information.

8. COMMENT: In Figure2 and 3, can the authors comment on why the same trends of Ca2+ release were not observed in the HUVEC cells but are conserved in the SHSY-5Y?

RESPONSE: With regard to the results of Figs 2. and 3., we consider that remimazolam caused a dose-dependent elevation of intracellular calcium in both SHSY-5Y cells and HUVECs. We consider that the tendencies of calcium elevation seen in SHSY-5Y cells and HUVECs were same both in the presence or absence of extracellular calcium. In order to avoid confusion, we have revised the descriptions concerning this point. (page 12, line 209-210)

9. COMMENT: Figure 8 A&B the data suggests that dantrolene was not providing a complete block of the RyR. Tetracaine would be a more efficient inhibitor.

RESPONSE: In accordance with the reviewer’s comment, we investigated the effects of tetracaine on remimazolam-induced intracellular calcium elevation. As shown in a Supplemental Figure.3, tetracaine also did not significantly affect the calcium elevation, although it significantly inhibited caffeine (10 mM)-induced calcium elevation. We have shown these data as Supplemental Figure.3 in revised manuscript.

Minor Points

10. COMMENT: Histogram formats should be changed to include individual data points.

RESPONSE: We were suggested the similar question by reviewer #2, so we modified the figures using the box plot.

11. COMMENT: Figure 1, states “the rate” of calcium, is this a typo that should state ratio as the authors have not measured the rate of Ca2+ release/increase?

RESPONSE: As the reviewer pointed out, we have changed the title of the figure to "a/b indicates the ratio of calcium elevation".

12. COMMENT: Figure 4, it appears the data panel of HUVECS with 500uM remimazolam is absent, was the data unable to be gathered under these conditions?

RESPONSE: As the reviewer mentioned, we could not collect data under these conditions.

 

GENERAL COMMENTS TO THE REVIEWER #2

RESPONSES TO THE REVIEWER #2’s COMMENTS:

I appreciate your very thoughtful comments as a reviewer.

*Specific Comments

1. COMMENT: Abstract:

Briefly mention that remimazolam causes an elevation in intracellular calcium and why this may be of interest.

RESPONSE: Many anesthetics, including Propofol, have been reported to induce elevation of intracellular calcium, and we were interested to investigate the possible contribution of calcium elevation to the mechanism of the newly approved remimazolam actions.

We added the above sentence to the beginning of the Abstract.

2. COMMENT: Introduction:

Line 75. Propose a mechanism by which propofol (or remimazolam) could cause hypotension or vascular pain. This mechanism could be used to justify the study of calcium mobilization and the use of neuronal-like cells and endothelial cells.

RESPONSE: Propofol-induced calcium elevation may induce vasodilation by activating the intracellular signaling pathway and promoting the phosphorylation of NO synthase, resulting in the synthesis of NO. This may contribute to hypotension and vascular pain.

We added the above sentence to the middle of the Introduction. （page 5, line 87 – 90）

3. COMMENT: Methods:

Lines 126-130. For reproducibility purposes, please provide more details about the apparatus and software used to measure calcium signals. Was a photodiode, PMT, or camera used (include manufacturer and model)? Also, add details about excitation light source, excitation, and emission filters, and objective used. What was the sampling rate?

RESPONSE: The observing images obtained by a BZ-9000 fluorescent microscope (KEYENCE) were used for the analysis. We did not use any software, photodiodes, or PMT. In this experiment, the excitation light source of a conventional fluorescence microscope was used. We used GFP-B (KEYENCE, excitation wavelength 470/40 nm

, emission wavelength 535/50 nm) as the excitation and emission filters and S Fluor X40/0.93 (NIKON) as the objective lens. Sampling rate was every 15 seconds. These are described in the Material & Method (page 9, line 140-142).

4. COMMENT: Statistical: Please clarify whether the normality test was assessed and how. When reporting results (significant or non-significant), please give the exact p-value, adjusted significance level (when appropriate), and the test used.

RESPONSE: In the revised paper, the statistical analysis method was changed to a nonparametric test since some studies had a small number of cases. Specifically, we used the Mann-Whitney test and described the exact P value. The description of the text and figure legends were changed in accordance with this revision.

5. COMMENT: Results:

Lines 194-204. “Observation of remimazolam…” Delete this section or show the results, negative results still a result. Data could be show as supplementary information.

RESPONSE: We followed the reviewers' comments and presented a new figure as supplemental figure.2.

6. COMMENT: To further characterize the involvement of GPCR, the authors could use GDPbetaS, to prevent non-specific activation of GPCR. If remimazolam effect depend on GPCR activation GDPbetaS should prevent its effects.

RESPONSE: GDPbetaS is difficult to use in our experimental system due to the fact that it is an inhibitor that does not penetrate cell membranes; GDPbetaS is commonly used in electrophysiological experiments by filling it into a pipette.

7. COMMENT: Discussion:

Lines 262. “ a newly marketed” delete this, it has been already mention few lines above. The authors wrote: “Remimazolam concentrations used in this study were considerably higher than those used clinically.”

RESPONSE: As you noted, we have removed it from our text.

8. COMMENT: You could add something like: However, concentrations like the ones used in this study or higher could occur locally near the injection site.

RESPONSE: Thank you for pointing this out. We have added the description you suggested to the Discussion (page 25, line 462-463).

9. COMMENT: It will be worth mentioning somewhere in the discussion that while remimazolam does not seems to affect RyR1 directly, the IP3-dependent elevation in Ca2+ mediated by remimazloam could play a role in sensitizing RyR1 in an individual with HM-susceptibility.

RESPONSE: Thank you for pointing this out. We have added the description you suggested to the Discussion. In addition, since it has been reported that there was no change in the sensitivity of intracellular calcium elevation to remimazolam in cells transfected with the MH mutation. Subsequently, we added the following statement.

“A recent study (Watanabe T et al. Biomed Res Int. 2021;2021:8845129) reported that there were no changes in the sensitivity to intracellular calcium elevation of remimazolam in cells transfected with the MH mutation.” (page 23-24,line 440-445).

10. COMMENT: Figure 1:

Traces in panel B appear identical to those previously published by the authors (see Ref #6 Urabe T et al. 2020, Fig. 1B). This issue is very problematic and needs to be fixed or request permission to reuse already published data. Also, in the same panel, how comes that the background signal is not flat. Would you please indicate the region of interest used to measure both the cell and background on the image?

RESPONSE: We apologize for confused Figure.1B. Figure.1B was modified to fit the data of this study. The areas where the background was taken are indicated by circles. We have chosen areas that appear as flat as possible, but some variation is allowed. The region of interest is the full screen.

11. COMMENT: Figures 2, 3, 4, 6-8.

Use box plot with data overlap instead of plunger bars (show the data).

Figures 2 and 3 add the number of observations to boxes as in figures 4,6-8.

RESPONSE: As a reviewer pointed out, we redesigned the figure using the box plot. In addition, I have added the number of observations to the box in Figures 2 and 3.

Minor Comments

12. COMMENT: Figure 2-8. Adding labels to the figure panels (i.e., Control solution for Fig 2, Zero Ca2+ for figure 3) may help the reader quickly identify the results.

RESPONSE: We have made the changes as the reviewer suggested as far as possible.

13. COMMENT: Figure 3A, remove data for DMSO 0.05-0.3 as in panel B.

RESPONSE: We have made the changes as the reviewer suggested. We also made the same changes in Figure 2 A. and B.

14. COMMENT: Figure 4. Increase Font size for BAPTA-AM and symbols

RESPONSE: We have made the changes as the reviewer suggested.

15. COMMENT: Figures 6-8. Increase the font size for x- and y-axis labels

RESPONSE: We have made the changes as the reviewer suggested.

---

## [Decision Letter · Decision Letter 1]

27 Dec 2021

PONE-D-21-15802R1Characterization of intracellular calcium mobilization induced by remimazolam, a newly approved intravenous anestheticPLOS ONE

Dear Dr. Miyoshi,

Thank you for submitting your manuscript to PLOS ONE. After careful consideration, we feel that it has merit but does not fully meet PLOS ONE’s publication criteria as it currently stands. Therefore, we invite you to submit a revised version of the manuscript that addresses the points raised during the review process. In line with the the comments of Reviewer#1, please redraw all figures where necessary using box plots.

We look forward to receiving your revised manuscript.

Kind regards,

Laszlo Csernoch, Ph.D.

Academic Editor

PLOS ONE

Journal Requirements:

Reviewers' comments:

Reviewer's Responses to Questions

**Comments to the Author**

1. If the authors have adequately addressed your comments raised in a previous round of review and you feel that this manuscript is now acceptable for publication, you may indicate that here to bypass the “Comments to the Author” section, enter your conflict of interest statement in the “Confidential to Editor” section, and submit your "Accept" recommendation.

Reviewer #1: (No Response)

Reviewer #2: All comments have been addressed

2. Is the manuscript technically sound, and do the data support the conclusions?

Reviewer #1: Yes

Reviewer #2: Yes

3. Has the statistical analysis been performed appropriately and rigorously? 

Reviewer #1: Yes

Reviewer #2: Yes

4. Have the authors made all data underlying the findings in their manuscript fully available?

Reviewer #1: Yes

Reviewer #2: Yes

5. Is the manuscript presented in an intelligible fashion and written in standard English?

Reviewer #1: Yes

Reviewer #2: Yes

6. Review Comments to the Author

Reviewer #1: The authors have done much to address reviewer comments and improve the manuscript.

However, Figures 1 and 2 are still displayed as summary histograms and have not been altered to show the spread of the individual data points. While the subsequent use of box plots is a positive adjustment, as the data is stated in the figure legends as mean and SEM this is still are not showing the spread of the data. The box plots could be altered to include the individual data points or to show mean with SD.

Reviewer #2: The authors have addressed all my queries, I have no further comments. I find the authors' responses and the revised manuscript satisfactory and I congratulate the authors for the work done.

7. PLOS authors have the option to publish the peer review history of their article (what does this mean?). If published, this will include your full peer review and any attached files.

Reviewer #1: No

Reviewer #2: **Yes: **Erick O. Hernández-Ochoa

---

## [Author Response · Author response to Decision Letter 1]

6 Jan 2022

GENERAL COMMENTS TO THE REVIEWER #1

RESPONSES TO THE REVIEWER #1’s COMMENTS:

1. COMMENT: The authors have done much to address reviewer comments and improve the manuscript. However, Figures 1 and 2 are still displayed as summary histograms and have not been altered to show the spread of the individual data points. While the subsequent use of box plots is a positive adjustment, as the data is stated in the figure legends as mean and SEM this is still are not showing the spread of the data. The box plots could be altered to include the individual data points or to show mean with SD.

RESPONSE: In accordance with the reviewer#1's suggestion, the histograms in figures 2 and 3 have been changed to box plots. We have also removed the description concerning mean and SEM in figure legends. And, we added the following statements: "The horizontal line in each box indicates the median, the box shows the interquartile range (IQR), and the whiskers represent 1.5 × IQR." The descriptions in Figure legend 4, 6, 7, 8 have been changed as well.

---

## [Editor Report · Decision Letter 2]

19 Jan 2022

Characterization of intracellular calcium mobilization induced by remimazolam, a newly approved intravenous anesthetic

PONE-D-21-15802R2

Dear Dr. Miyoshi,

We’re pleased to inform you that your manuscript has been judged scientifically suitable for publication and will be formally accepted for publication once it meets all outstanding technical requirements.

Kind regards,

Laszlo Csernoch, Ph.D.

Academic Editor

PLOS ONE
---

## [Editor Report · Acceptance letter]

21 Jan 2022

PONE-D-21-15802R2 

Characterization of intracellular calcium mobilization induced by remimazolam, a newly approved intravenous anesthetic 

Dear Dr. Miyoshi:

I'm pleased to inform you that your manuscript has been deemed suitable for publication in PLOS ONE. Congratulations! Your manuscript is now with our production department. 

Kind regards, 

on behalf of

Dr. Laszlo Csernoch 

Academic Editor

PLOS ONE